# MicroRNAs as Key Players in Melanoma Cell Resistance to MAPK and Immune Checkpoint Inhibitors

**DOI:** 10.3390/ijms21124544

**Published:** 2020-06-26

**Authors:** Maria Letizia Motti, Michele Minopoli, Gioconda Di Carluccio, Paolo Antonio Ascierto, Maria Vincenza Carriero

**Affiliations:** 1Department of Motor and Wellness Sciences, University “Parthenope”, 80133 Naples, Italy; 2Neoplastic Progression Unit, Istituto Nazionale Tumori IRCCS ‘Fondazione G. Pascale’, 80131 Naples, Italy; m.minopoli@istitutotumori.na.it (M.M.); g.dicarluccio@istitutotumori.na.it (G.D.C.); 3Melanoma, Cancer Immunotherapy and Development Therapeutics Unit, Istituto Nazionale Tumori-IRCCS Fondazione “G. Pascale”, 80131 Naples, Italy; p.ascierto@istitutotumori.na.it

**Keywords:** miRNA, melanoma, melanoma resistance to MAPK/MEK inhibitors, resistance to immune checkpoint inhibitors

## Abstract

Advances in the use of targeted and immune therapies have revolutionized the clinical management of melanoma patients, prolonging significantly their overall and progression-free survival. However, both targeted and immune therapies suffer limitations due to genetic mutations and epigenetic modifications, which determine a great heterogeneity and phenotypic plasticity of melanoma cells. Acquired resistance of melanoma patients to inhibitors of BRAF (BRAFi) and MEK (MEKi), which block the mitogen-activated protein kinase (MAPK) pathway, limits their prolonged use. On the other hand, immune checkpoint inhibitors improve the outcomes of patients in only a subset of them and the molecular mechanisms underlying lack of responses are under investigation. There is growing evidence that altered expression levels of microRNAs (miRNA)s induce drug-resistance in tumor cells and that restoring normal expression of dysregulated miRNAs may re-establish drug sensitivity. However, the relationship between specific miRNA signatures and acquired resistance of melanoma to MAPK and immune checkpoint inhibitors is still limited and not fully elucidated. In this review, we provide an updated overview of how miRNAs induce resistance or restore melanoma cell sensitivity to mitogen-activated protein kinase inhibitors (MAPKi) as well as on the relationship existing between miRNAs and immune evasion by melanoma cell resistant to MAPKi.

## 1. Introduction

Melanoma represents one of the most aggressive skin cancers with a significantly increased incidence in the last decades [1,2,3]. Currently, therapeutic options include surgical excision, chemotherapy, targeted and immune therapies administered as single agents or in combination, depending on the stage of the disease, location, as well as the genetic profile of the tumor [4]. In the last years, molecular targeted therapies and immunotherapies have significantly improved the overall survival of patients with metastatic disease [5,6].

In the past years, either dabrafenib or vemurafenib BRAF inhibitors (BRAFi) showed encouraging response rates, although the duration of response appeared to be limited [7,8]. BRAF inhibitor resistance depends on oncogenic signaling through reactivation of MAPK/Erk or activation of PI3K/Akt, which may be acquired by directly affecting genes in each pathway, by upregulation of receptor tyrosine kinases, or by affecting downstream signaling [9]. Thus, the combination of dabrafenib with the MEK inhibitor (MEKi) trametinib, has become employed worldwide for the care of patients with BRAF-mutant metastatic melanoma, improving their progression-free and overall survival [10,11]. Unfortunately, patients treated with dabrafenib/trametinib combination therapy also develop alterations in the same genes that support single-agent resistance including MEK1/2 mutations, BRAF amplification, BRAF alternative splicing, and NRAS mutations [12,13]. The limiting factor for these therapeutic approaches is the heterogeneity and phenotypic plasticity of melanoma cells due to genetic mutations and epigenetic modifications that may determine the paradoxical activation of the mitogen-activated protein kinase (MAPK) and thus sustain resistance to these drugs [14]. The new immune checkpoint blockade therapies improve the outcomes of patients with advanced melanoma regardless of the mutation status and several ongoing clinical trials highlight that combinations of BRAFi and MEKi with immune checkpoint inhibitors result in more durable responses in about 50% of patients [15,16,17]. Based on these considerations, the identification of biomarkers that monitor and/or predict an early response during melanoma therapy still represents an unmet clinical need.

Using a variety of technical approaches such as chromosomal analysis, miRNA microarrays, miRNA qPCR arrays, and high-throughput small RNA sequencing platforms, microRNA (miRNA)s have been identified to function as oncogenes or tumor repressors genes. Oncogenic miRNAs (oncomiRs) are frequently overexpressed in cancers while tumor-suppressive miRNAs are down-regulated. It has been documented that miRNAs regulate more than 30% of human protein-coding genes [18] and control, through degradation of mRNA or a translation block, numerous cancer-relevant processes including proliferation, autophagy, migration, and apoptosis [19]. Specific miRNA signatures have been found differentially expressed in normal and tumor tissues, suggesting their potential value as molecular biomarkers useful for diagnosis, staging, progression, prognosis, and response to treatments [20,21,22].

miRNAs are short, single-stranded, non-coding nucleotide sequences with an average 22 nucleotides in length. They are transcribed as individual genes, from introns of coding genes (intronic miRNAs) or from regions between the clusters of genes (intergenic miRNAs) while clustered miRNAs are transcribed as polycistronic transcripts [23]. miRNA genes are transcribed by RNA polymerase II into primary miRNAs (pri-miRNA)s, processed into precursor miRNA’s (pre-miRNA)s and then into mature miRNAs. After processing, mature single-stranded miRNAs associate with argonaute protein family (Argo) and glycine-tryptophan proteins of 182 kDa (GW182), which are the principal constituents of the miRNA-induced silencing complex (miRISC) [24], and usually bind to the 3′UTRs of their cytosolic mRNA targets, resulting in mRNA-reduced translation or deadenylation and degradation of the mRNA transcript [25]. The interaction of miRNAs with other regions, including the 5′UTR coding sequence, and gene promoters, has also been reported [26,27]. miRNA interaction with target genes may be influenced by several factors, including the subcellular location of miRNAs, abundancy of miRNAs and/or corresponding target mRNAs, as well as the affinity of miRNA-mRNA interactions [28]. Moreover, recent studies suggest that miRNAs may be shuttled between different subcellular compartments to control the rate of translation and transcription [29] and that an individual miRNA can act on several mRNA simultaneously, modulating multiple processes in cancer cells in a cooperative manner [30,31]. Furthermore, some microRNAs are related to the expression of transmembrane oncogenes, acting directly on their expression (e.g., EGFR) [32], or acting indirectly, by regulating the expression of soluble ligands that recognize specifically the extracellular domain of the receptors [33].

It has been shown that chromosomal rearrangements, epigenetic regulation and disorders in miRNA biogenesis, result in increased or decreased expression of miRNAs in melanoma cells as compared to melanocytes [34,35,36]. Furthermore, miRNA altered expression has been described in different stages of melanoma progression, so that expression levels of specific miRNAs are considered as diagnostic and/or prognostic biomarkers in melanoma [37,38,39,40]. When secreted into extracellular fluids, miRNAs are stable in human fluids since they are packaged in exosomes and microvescicles or associated with RNA-binding proteins such as Argo2 or lipoprotein complexes, which protect them from degradation [28,41,42]. In a recent study, 11 miRNAs were identified as differentially expressed between healthy controls and plasma samples from different melanoma stages [43]. Therefore, miRNAs, especially those being part of the circulating transcriptome, may be useful as biomarkers for early melanoma detection and response to treatments [44]. Numerous miRNAs have been found to regulate melanoma cell behavior and gene expression acting on the MAPK signaling pathway [45], while some miRNAs have been found to regulate the expression of immune checkpoints, acting on melanoma cells or immune cells [46].

In this review, we discuss the latest progress regarding mechanisms by which miRNAs regulate melanoma cell resistance to MAPKi and immune evasion. Furthermore, the potential predictive value of circulating miRNAs for monitoring melanoma responsiveness to targeted and immune therapies is debated.

## 2. miRNAs Involved in the Regulation of Melanoma MAPKi-Resistance

In recent years, by next-generation sequencing, the Cancer Genome Atlas provided the analysis on the somatic aberrations underlying melanoma genesis, identifying BRAF, RAS, and NF1 mutant genetic subtypes of cutaneous melanoma, all of them being able to deregulate the MAPK/ERK pathway, leading to uncontrolled cell growth [47]. Over 50% of melanomas harbor activating mutation in the BRAF gene, which sustains proliferation and survival of melanoma cells by activating the MAPK pathway. Over 90% BRAF mutations are at codon 600 and among these, over 90% are a single nucleotide mutation resulting in substitution of the valine with a glutamic acid residue (BRAFV600E), while less common mutations are the substitutions of valine with lysine, arginine, leucine or aspartic acid residues [48]. Vemurafenib and dabrafenib BRAF inhibitors (BRAFi) have improved the outcomes of patients with BRAF-mutant metastatic melanoma [7,8]. Unfortunately, most of them develop drug resistance early as a consequence of the activation of alternative proliferation-inducing pathways, often associated to the reactivation of the MAPK pathway [49,50,51,52,53]. Indeed, resistance also occurs in the majority of melanoma patients treated with BRAFi and MEKi combinations, although overall and progression-free survival are prolonged compared to single-agent therapies [54,55]. Furthermore, it has to be taken into account that BRAF-mutant melanomas may acquire BRAF inhibitor resistance via up-regulation of both MAPK and PI3K/Akt pathways in about 22% of the melanoma patients [49], whereas other drugs targeting different cellular pathways may escape development of drug resistance, probably due to the extraordinary plasticity of melanoma cells [56,57,58].

During the progressive development of drug resistance, several deregulated miRNAs have been shown to control both tumor cell growth and melanoma cell interactions with the tumor microenvironment. Some miRNAs provoke drug resistance while others restore drug sensitivity. In Table 1, miRNAs with a potential role in regulating melanoma sensitivity and resistance to MAPKi and the underlying mechanisms of action are listed.

First, Liu and co-workers showed that miR-200c is a potential therapeutic target to restore melanoma cell sensitivity to BRAFi. They found that miR-200c reverts drug resistance to PLX4720 BRAF and U0126 MEK inhibitors by down-regulating the p16 transcriptional repressor BMI-1, which, in turn, inhibits melanoma cell growth and metastases in nude mice. Moreover, they found that miR-200c acts on ABC transporters, a superfamily of transmembrane proteins that mediate drug resistance in melanoma cells [59]. The clinical significance of miR-200c/Bmi1 axis in inhibiting acquired resistance to BRAFi was confirmed in human melanoma tissues: loss of miR-200c expression was found to correlate with development of resistance to BRAFi and promote the development of a BRAFi-resistant phenotype in melanoma cells and in melanoma tissues with a mechanism that involves MAPK and PI3K/AKT signaling pathways [60]. Like miR-200c, miR-524-5p expression appeared down-regulated in melanoma cells with activated MAPK/ERK pathway. miR-524-5p suppresses MAPK/ERK pathway-triggered melanoma cell proliferation by directly binding to the 3’-UTR of both BRAF and ERK2 [61]. Fattore L. and colleagues found that miR-579-3p is down-regulated in vemurafenib-resistant melanoma cells and that its ectopic expression impairs the establishment of drug resistance in human melanoma cells. They also showed that down-regulation of miR-579-3p in tumor tissues from melanoma patients with acquired resistance to BRAFi well correlates with a poor prognosis [62]. Mechanistically, miR-579-3p binds to the 3’UTR of either BRAF and MDM2, an E3 ubiquitin protein ligase that promotes p53 degradation [63], so that MDM2 and p53 cause a negative-feedback loop, in which p53 induces the expression of MDM2 [62]. The miR-506-514 cluster has been shown to regulate not only melanocyte transformation but also melanoma cell proliferation [64]. Stark and coworkers demonstrated that miR-514a, which is expressed in 69% of melanoma cell lines, reverts drug resistance to BRAFi by directly binding to NF1 transcripts, leading to altered NF1 protein expression and consequent decreased cell proliferation. Accordingly, overexpression of miR-514a increases survival of vemurafenib-treated BRAF(V600E) melanoma cells [65]. A microarray profiling analysis of vemurafenib-resistant and sensible A375 melanoma cells allowed Sun X. and colleagues to identify 17 dysregulated miRNAs in BRAFi resistant A375 cells. Among these, miR-7 was found to be the most down-regulated miRNA that prevents proliferation and partially reverts drug resistance of vemurafenib-resistant melanoma cells [32]. miR-7 inhibits both MAPK and PI3K/Akt signaling pathways by targeting EGFR, IGF-1R and CRAF [32]. In this regard, miR-7 could inhibit the activation of the MAPK and PI3K/AKT pathways and reverse melanoma cell resistance to BRAFi, by decreasing the expression levels of EGFR and IGF-1R. Using real time quantitative PCR and microarray analyses, Kim JA and co-workers found that up-regulation of miR-1246 associates with acquired resistance to BRAFi by A375P melanoma cells. Although the exact mechanism of action of miR-1246 in eliciting drug resistance has been not yet completely identified, Authors provided evidence that resistance to PLX4720 in miR-1246 mimic-transfected cells is mostly due to the inhibition of autophagy [66]. By miRNA expression profiling of sensible and BRAFi resistant melanoma cells, Lisa Koetz-Ploch and colleagues found that miR-125a becomes overexpressed upon acquisition of cell resistance to BRAFi. Mechanistically, miR-125a suppresses the apoptotic program in BRAFi-treated melanoma cells by targeting two components of the intrinsic pro-apoptotic pathway: BAK1 and MLK3 [67]. The finding that miR-125a is up-regulated in tissues of BRAFi-treated melanoma patients as compared to tumor samples excised before BRAF-treatment, allowed Authors to propose the use of anti-miR-125-a for preventing or overcame BRAFi resistance [67].

Melanoma cells are documented to release into the extracellular milieu different types of extracellular vesicles (EV)s, including oncosomes, ectosomes, exosomes, and melanosomes carrying protein and small RNAs cargos [68]. Comparing RNA sequences of exosomal miRNA released by a number of melanoma cell lines with clinical miRNA datasets from human melanoma tissue samples, Lunavat TR and coworkers found that the exosomal miR-214-3p, miR-199a-3p and miR-155-5p associate with melanoma progression [69]. More recently, the same Authors found that both vemurafenib and dabrafenib BRAFi significantly increase expression of miR-211-5p in EVs from melanoma cell cultures and tissues, leading to re-activation of the survival pathway. Mechanistically, overexpression of miR-211-5p depends on BRAFi-induced up-regulation of the microphthalmia-associated transcription factor (MITF) which, in turn, induces activation of the survival pathway trough the master regulator TRPM1 gene [70]. By carrying out RNA-seq analyses, Díaz-Martínez and co-workers documented in vemurafenib-resistant A375 cells very high levels of miR-204-5p and miR-211-5p when compared to parental counterparts. They found that, when engrafted in mice, sensible A375 cells transfected with both miR-204-5p and miR-211-5p became resistant to vemurafenib and were able to grow, whereas resistant cells silenced for miR-204-5p and miR-211-5p expression lost tumor growth ability and became sensible to vemurafenib [71]. Mechanistically, co-overexpression of miR-204-5p and miR-211-5p triggers Ras and MAPK up-regulation not only in response to BRAFi but also in response to inhibitors of other downstream effectors of the MAPK pathway [71]. Examination of some potential targets for these miRNAs revealed that miR-204-5p or miR-211-5p reduce significantly at mRNA and protein levels the NUAK1/ARK5 kinase [71]. Accordingly, NUAK1/ARK5 protein was consistently reduced in vemurafenib-resistant cells [71].

Overexpression of the Yes-associated protein (YAP) has been found to associate with resistance to anticancer therapies in solid tumors, including BRAFi resistant melanomas [72,73]. miR-550a-3-5p overexpression has been proven to down-regulate YAP at mRNA and protein levels and YAP down-regulation-dependent tumor-suppressive activity induces sensitization of BRAFi-resistant melanoma cells to vemurafenib [74]. Fattore L. and coworkers demonstrated that down-modulation of miR-199b-5p in drug-resistant melanoma cells causes increased VEGF release and acquisition of a pro-angiogenic status that may be reverted by restoring miR-199b-5p levels [33]. In line with these findings, the occurrence of a miRNA-dependent regulation of VEGF production in melanoma cells resistant to BRAF inhibitors was documented by Caporali and colleagues [75]. These Authors found low levels of miR-126-3p in dabrafenib-resistant melanoma cells as compared with their parental counterparts and that proliferation and invasiveness of dabrafenib-resistant cells may be reduced by restoring the miR-126-3p expression [75]. By analyzing the global miRNAome changes in sensible and BRAFi-resistant melanoma cells, Fattore L. et colleagues identified many deregulated miRNAs involved in the acquisition of drug resistance to BRAFi. They identified specific miRNA signatures capable of distinguishing drug responding from non-responding patients as well as a subset of miRNAs capable to block or revert the development of drug resistance when down- or up-regulated. Using qRT-PCR on matched tumor biopsies and serum samples from melanoma patients, the same Authors found that miR-204-5p and miR-199b-5p are down-regulated in relapsing melanomas, whereas miR-4443 and miR-4488 are up-regulated [33]. Accordingly, they found that overexpression of down-regulated miR-204-5p and miR-199b-5p reduces cell proliferation and induces apoptosis, whereas inhibition of up-regulated miR-4443 and miR-4488 with specific antagomiRs, restores inhibitory effects exerted by BRAFi. Authors also found that a reduced proliferation of A375 melanoma cells double resistant to BRAFi and MEKi, may be achieved by down-regulating simultaneously miR-204-5p, miR-199b-5p and miR-579-3p, highlighting the notion that co-targeting multiple microRNAs may be a valid approach to prevent proliferation of melanoma cells with acquired resistance to BRAFi and MEKi [33].

**Table 1 ijms-21-04544-t001:** microRNAs Involved in the Acquisition of Melanoma Cell Resistance to MAPK Inhibitors.

miRNAs	Expression	Target Gene/s	Mechanism/s	Tissue/Cell Lines/Blood	Reference
miR-200c	Down	BMI1, ZEB2, TUBB3, ABCG5, MDR1	p16 Transcriptional Repressor BMI-1/up-Regulation of ABC Transporters. Activation of MAPK and PI3K/AKT Signaling Cascades	Tissues, Cell lines	[59,60]
miR-579-3p	Down	BRAF and MDM2	Reduced Proliferation (by Targeting BRAF). Increased Apoptosis (by Down-Regulating of MDM2)	Tissues, Cell lines	[62]
miR-7	Down	EGFR, IGF-1R, CRAF	Inhibition of MAPK and PI3K/Akt Signaling Pathways	Cell lines	[32]
miR-550a-3-5p	Down	YAP	Reduced Proliferation through YAP Inhibition	Cell lines	[74]
miR-199b-5p	Down	HIF-1α, VEGFA	Pro-Angiogenic Activity	Tissues, Cell Lines, Plasma	[33]
miR-126-3p	Down	VEGFA, ADAM9	Increased Proliferation through the p-ERK1/2, p-Akt//VEGF axis	Cell Lines	[75]
miR-204-5p, miR-199b-5p	Down	BCL-2, FOXM1, NOTCH, VEGF	Increased Survival/Reduced Apoptosis Bcl2, HIF-1/VEGF	Tissues, Cell Lines, Plasma	[33]
miR-514a	Up	NF1	Inhibition of NF1 Increased Survival	Cell Lines	[65]
miR-1246	Up	NS	Inhibition of Autophagy	Cell Lines	[66]
miR-125a	Up	BAK1 and MLK3	Inhibition of Apoptotic Program	Tissues, Cell Lines	[67]
miR-204-5p, miR-211-5p	Up	NUAK	Up-Regulation of the Ras/MEK/ERK Pathway through MITF/Increased Survival Pathway	Tissues, Cell Lines	[70,71]
miR-4443, miR-4488	Up	Autophagy-Related Genes	Deregulation of Autophagy	Tissues, Cell Lines, Plasma	[33]

List of miRNAs involved in the melanoma resistance to MAPK inhibitors. Up/Down expression levels are referred to resistant melanoma cells. Not shown (NS) indicates that miRNAs target genes have not been identified in the corresponding studies.

## 3. miRNA in Melanoma Cell Resistance to Immunotherapy

There are several attempts to investigate the potential link between miRNAs expression profile and patients’ response to immune checkpoint inhibitors in order to verify at the same time their potential use for monitoring efficacy of immune checkpoint blockade and improving the outcomes of patients with advanced melanoma. Although few data regarding miRNAs and immune checkpoint inhibitors relationship are available in the literature, recent studies demonstrate that some miRNAs may regulate directly or indirectly the expression of immune checkpoints, acting on tumor cells or immune cells, respectively (Table 2).

Galore-Haskel and collaborators found higher levels of miR-222 in melanoma tissues from patients that were non-responders to ipilimumab when compared to responder patients, raising the possibility that miR-222 expression could be considered a valid biomarker for predicting responsiveness of melanoma patients to ipilimumab [76]. These Authors documented that Adenosine Deaminase Acting on RNA-1 (ADAR1) overcomes melanoma immune resistance and increase proliferation of melanoma cells by regulating the biogenesis of miR-222 at transcriptional level [76]. miR-222 directly interact with 3’UTR of the Intracellular Adhesion Molecule 1 (ICAM1) mRNA [77] which, consequently, affects melanoma immune resistance by rendering melanoma cells more resistant to TIL-mediated killing mainly due to their ability to cross endothelial vessels and infiltrate tumor tissues [78,79]. Analyzing exosomal miRNAs in sera from melanoma patients, Tengda and co-workers found higher levels of miR-532-5p and miR-106b in melanoma patients with stage III–IV disease, as compared to patients with stage I–II disease and low levels of miR-532-5p and miR-106b in melanoma patients treated with pembrolizumab compared to those untreated. The Authors concluded that measurement of exosomal miRNA-532-5p and miRNA-106b in the sera from melanoma patients could be used for monitoring and/or predicting their response to immunotherapies [80].

miRNAs are also involved in the regulation of immune cells within the tumor microenvironment, including cytotoxic, CD4 or γδ T lymphocytes, natural killer (NK), macrophages and myeloid-derived suppressor cells (MDSCs).

A direct involvement of tumor-suppressor miRNAs in the control of antitumor immune response through the regulation of immune checkpoints PD-1, PD-L1, and CTLA-4 has been ascertained in tumors of different origin [81]. By a microarray-based profiling performed in PD1+ and PD1- CD4 T cells sorted from lymph nodes and spleen of melanoma-bearing mice, Li and colleagues demonstrated that miR-28 decreases PD1 expression by directly binding to its 3’UTR, suggesting that miR-28 regulates exhaustive differentiation of Treg in melanoma cells. Moreover, exhausted T cells showed a reduced secretion of IL-2, TNF-α and IFN-γ and the use of miR-28 mimics was able to restore their secretion [82]. Martinez-Usatorre and co-workers analyzed miR-155 expression in CD8^+^ T cells isolated from tumor-infiltrated lymph nodes and tumor tissues of melanoma patients and murine models. They found that miR-155 up-regulation within the tumors correlates with increased CD8^+^ T-cell infiltration while low expression of miR-155 targets in melanoma tumors associates with a prolonged overall survival. These findings allowed Authors to conclude that miR-155 could be considered a marker of responsiveness of CD8 T cells, as further demonstrated by its up-regulation after PD1 blockade [83].

Up-regulation of stress-induced ligands, including ULBP2, allows tumor cell recognition by immune cells trough the NKG2D receptor expressed on lymphocytes, Natural Killer cells, as well as cytotoxic, CD4 or γδ T cells [84]. miR-34a and miR-34c have been shown to enhance NK-cell killing activity against melanoma cells by targeting the UL16 binding protein 2, while miR-34 mimics led to down-regulation of ULBP2, diminishing tumor cell recognition by NK cells [85]. By using next-generation sequencing, Cobos JV and colleagues identified a repertoire of miRNAs that have a specific expression signature in M2 polarized macrophages [86]. A panel of miRNAs have been recognized to promote the conversion of monocytes into myeloid-derived suppressor cells (MDSC)s, their baseline levels being found to correlate with the clinical efficacy of immune checkpoint inhibitors. For instance, miR-125a-5p inhibits M1 polarization and promotes the alternative M2 phenotype by targeting KLF13, a transcriptional factor that is active during T lymphocyte activation [87,88]. Moreover, both miR-146a and miR-146b promote M2 polarization in human and mouse models by down-regulating pro-inflammatory responses [88]. Finally, several circulating miRNAs (let-7e, miR-99b, miR-100, miR-125a, miR-125b, miR-146a, miR-146b, and miR-155) were found to correlate with a shorter progression free and overall survival in melanoma patients treated with ipilimumab and nivolumab, thus representing the first predictive peripheral blood biomarker of resistance to immune checkpoint inhibitors [46]. These miRNAs released in the blood by melanoma EVs act by converting monocytes into MDSC and reduce the clinical efficacy of the PD-1 and CTLA-4 inhibitors [46]. Based on these findings, it will be foreseeing that combinations of miRNAs with different immune checkpoint targets could mimic or improve the effect of immune checkpoint blockade therapies.

## 4. Relationship between miRNAs and Immune Evasion by Melanoma Cell Resistant to MAPKi

The activation of the MAPK pathway through BRAF mutations leads to downstream production of several cytokines that promote tumor growth and immune evasion with autocrine or paracrine mechanisms. Recent studies have documented that the MAPK signaling pathway may be considered as a potential molecular target for overcoming melanoma cell evasion of the immune surveillance (Figure 1). By activating the MAPK cascade, the BRAF(V600E) mutation stimulates melanoma cells to produce a wide spectrum of chemokines and cytokines which, in turn, are responsible for the recruitment of immune and myeloid cells. For the first time, Sumimoto H. and co-workers, using the U0126 MEK inhibitor and lentiviral BRAF(V600E) RNA interference, found that the oncogenic BRAF favors melanoma immune escape increasing production of IL-6 and IL-10 which increase T-cell stimulatory function of dendritic cells [89]. Furthermore, constitutively activated BRAF(V600E) in melanoma tumor cells has been shown to initiate and sustain IL-1α/β-dependent T-cell suppression in a murine model. Mechanistically, IL-1α and IL-1β secreted by melanoma cells increase COX-2, PD-L1, and PD-L2 expression levels in tumor associated fibroblasts which, in turn, suppress the function of tumor-infiltrating T cells [90]. Jiang X. and colleagues identified the molecular mechanism by which melanoma cells resistant to BRAFi can evade the immune system via PDL-1 up-regulation. By using a panel of melanoma cell lines harboring BRAF(V600E) mutation, the Authors showed that the BRAFi resistance leads to c-Jun and STAT3-mediated increase of PD-L1 expression [91]. Conversely, the same Authors demonstrated, in vitro, that the U0126 MEK inhibitor simultaneously counteracts MAPK reactivation and reduces PD-L1 expression [91]. Analyzing several melanoma cell lines resistant to BRAFi as well as plasma and tumor samples from vemurafenib-treated melanoma patients, Vergani and coauthors found that BRAFi-resistant melanoma cells secrete higher levels of CC-chemokine ligand 2 (CCL2) then sensible counterparts. The CCL2 increase elicits up-regulation of miR-34a, miR-100 and miR-125b, which, in turn, down-regulate the canonical genetic pathway for apoptosis. Conversely, down-regulation of CCL2 and/or miR-34a restores apoptosis and melanoma sensitivity to vemurafenib [92]. More recently, miRNAs have been directly associated with melanoma resistance to treatment with immune checkpoint inhibitors (Figure 1). Audrito V. and coworkers found that PD-L1 expression is limited to a subset of patients with metastatic melanoma and unfavorable prognosis [93]. These Authors found that resistance to BRAFi and MEKi associates with induction of PD-L1 expression in BRAF(V600E)-mutated melanoma cell lines and identified the post-transcriptional circuit responsible for PD-L1 up-regulation, consisting of a direct interaction of miR-17-5p with the 3’UTR mRNA of PD-L1 [93]. Finally, miR-17-5p levels were found to inversely correlate with PD-L1 expression and thus predict sensitivity to BRAFi in patients with metastatic melanoma [93]. In this contest, modulating miRNAs impinging both MAPK pathway and immune responses could be a useful approach for treating patients with advanced melanoma.

## 5. Predictive Value of Circulating miRNAs for Monitoring Melanoma Responsiveness to Targeted and Immune Therapies

To date, there is an urgent need to develop new non-invasive methods for monitoring disease progression or resistance to treatments of melanoma patients. In this regard, liquid biopsy may be considered a non-invasive source of biomarkers, potentially useful for monitoring responsiveness of melanoma patients to targeted and immune therapies, although the strategies for these approaches are still under investigation. In the last decade, many efforts have been made to identify diagnostic and prognostic circulating miRNA biomarkers for melanoma. Circulating miRNAs have emerged as powerful biomarkers since they are highly stable in body fluids, which are protected against enzymatic degradation thanks to their association with RNA binding proteins (Argonaute-2 and nucleophosmin-1), with high- and low-density lipoproteins, or to their embedding in membrane vesicles such as exosomes [28,41,42]. Also, they are resistant to both high or low pH, multiple freeze-thaw cycles, and long-term storage [94]. In a recent review article, Gajos-Michniewicz A summarizes studies reporting significant alterations in the miRNA expression profile in the serum and plasma of melanoma patients compared to healthy controls, suggesting circulating miRNAs as promising diagnostic melanoma biomarkers [40]. In a recent study, Solé C and colleagues found 11 miRNAs (let-7b, miR-16, miR-21, miR-92b, miR-98, miR-134, miR-320a, miR-486, miR-628, miR1180, and miR-1827) that are differentially expressed between healthy controls and plasma samples from different melanoma stages [43].

As above described, numerous miRNAs have been shown to modulate melanoma sensitivity and resistance to MAPKi (Table 1) and/or immune checkpoint inhibitors (Table 2). Among these, some miRNAs are present not only in tissue samples but also in serum or plasma of melanoma patients, thus representing soluble putative markers to monitor the therapeutic responses to MAPKi and immune treatments. miR-199b-5p expression levels were found downregulated in the plasma of melanoma patients post-MAPKi treatment as compared to plasma from the untreated ones, whereas miR-4488 levels were significantly increased in patients after MAPKi treatment, indicating that these miRNAs may represent soluble putative markers to monitor the therapeutic responses to MAPKi [68]. Svedman and co-workers identified let-7g-5p and miR-497-5p as predictive biomarkers of MAPKi treatment benefit in metastatic melanoma patients. They analyzed miRNA content in the extracellular microvesicles recovered from plasma of melanoma patients before and after the treatment with MAPKi. Both let-7g-5p and miR-497-5p levels were found to increase after the treatment with MAPKi and to correlate with a prolonged progression-free survival [95]. By performing Nanostring nCounter analysis of 48 plasma samples from individuals with or without melanoma, Van Laar R and coworkers identified a set of thirty-eight independently validated circulating miRNAs. The so-called MEL38 signature includes some miRNA (hsa-miR-34a-5p, hsa-miR-299-3p, hsa-miR-624-3p, hsa-miR-1-5p, hsa-miR-152-3p, hsa-miR-1973, hsa-miR-454-3p, hsa-miR-4532) involved in the drug/immune resistance [96]. Eight miRNAs (let-7e, miR-99b, miR-100, miR-125a, miR-125b, miR-146a, miR-146b, and miR-155) detected in patients receiving ipilimumab and nivolumab have been found to correlate with the frequency of altered myeloid cells, shorter progression-free survival as well as overall survival [46].

Finally, Tengda L. and coworkers demonstrated that miR-532-5p and miR-106b, isolated from serous exosomes as well as from total serum, were able to discriminate patients with melanoma from healthy controls, metastatic patients from those with no metastasis, patients with stage I–II disease from those with stage III–IV, and patients treated with pembrolizumab from untreated ones [80].

## 6. Conclusions and Future Perspectives

In melanoma, several miRNAs are deregulated because of epigenetic changes, impaired transcription, amplification, or deletion of miRNA genes as well as defects in the miRNA biogenesis machinery. It is currently accepted that distinct profiles of miRNA expression are detected at each step of melanoma development, and that an altered expression of miRNAs frequently correlates with poor prognosis and/or inadequate response to treatments. As recapitulated in this review, dysregulated miRNAs may induce and sustain or prevent melanoma cell resistance to BRAFi /MEKi and immune therapies by acting as oncogenes or tumor suppressors, respectively. A partial or complete reversion of melanoma cells resistance to BRAFi and MEKi may be achieved by restoring down-regulated miRNAs or silencing up-regulated miRNAs, suggesting that specific miRNAs or their antagonists may be considered for potential therapeutic applications to overcame melanoma cell resistance to BRAFi and MEKi. In this regard, miRNAs, especially those being part of the circulating transcriptome, may be useful as biomarkers for early melanoma response to treatments, but the strategies for these approaches are still under investigation. In melanoma, implications of microRNAs in the regulation of immune checkpoint blockade and controlling their expression for therapeutic purposes is the subject of intense ongoing research. Specific miRNA signatures associate with specific alterations of immune checkpoint pathways in the melanoma microenvironment while subsets of miRNAs directly regulate the transcription of immune checkpoints. Thus, miRNA could provide new biomarkers predicting patient response to immune checkpoint inhibition and it is reasonable to foresee that combining miRNAs with different immune checkpoint targets could mimic and possibly improve the effect of combined immune checkpoint blockade therapies. Activation of the MAPK pathway through BRAF mutations may be a potential molecular target for overcoming evasion of the immune surveillance by melanoma cells. MAPK cascade stimulates melanoma cells to secrete cytokines, chemokines and soluble growth factors that recruit immune and myeloid cells sustaining both tumor growth and immune evasion. In this contest, specific miRNAs or their antagonists may be considered for potential therapeutic use for restoring the effector function of immune cells. New approaches that look at simultaneous or sequential use of drugs targeting the MAPK pathway with immune checkpoint inhibitors are also a priority, with evidence suggesting that specific miRNAs may overcome melanoma growth and immune evasion. Thus, many questions regarding the best first- and second-line treatment and the best treatment sequence remain to be addressed.

## Figures and Tables

**Figure 1 ijms-21-04544-f001:**
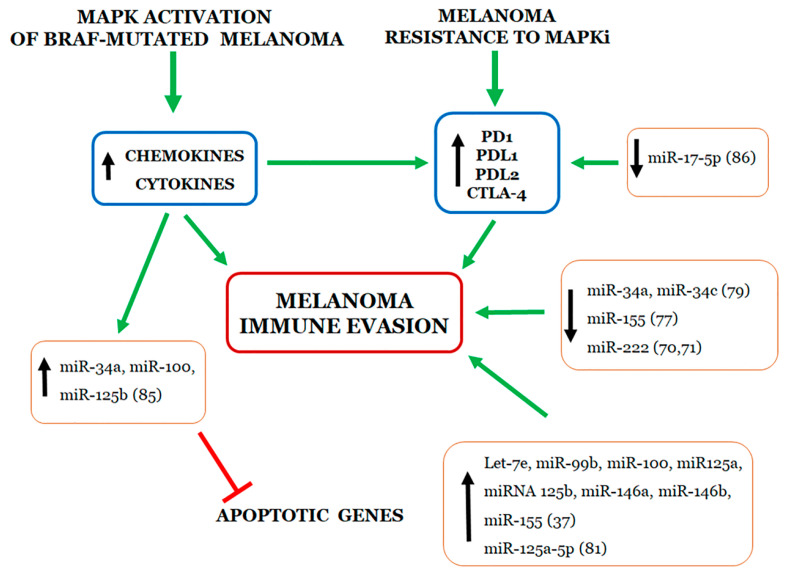
Schematic representation of up- (arrow pointing up) or down- (arrow pointing down) regulated miRNAs involved in evasion of immune surveillance by melanoma cells harboring BRAF mutations.

**Table 2 ijms-21-04544-t002:** microRNAs Involved in the Acquisition of Melanoma Resistance to Immune Checkpoint Inhibitors.

miRNA	Tissue/Cell Lines/Blood	Target/Function/Proposed Mechanism	Reference
miR-222	Tissues, Cell Lines	ADAR1/ICAM-Dependent -Increased Trans-Endothelial Migration of T Cells. Reduced Response to Ipilimumab	[76]
miR-532-5p, miR-106b	Serum Exosomes	Reduced Response to Pembrolizumab	[80]
miR-28	Cell Lines	Reduced PD1 Expression and Response to Pembrolizumab. Increased Differentiation of Treg. Reduced Secretion of IL-2, TNF-α and IFN-γ	[82]
miR-155	Tissues, Cell Lines, PBMC	Increased CD8+ T-Cell Infiltration	[83]
miR-34a, miR-34c	Cell Lines	Target UL16 Binding Protein 2I (ULBP2). Increased NK-cell Killing Activity	[85]
miR-125a-5p	Cell Lines	Targets KLF13. Protumoral Activity Trough Macrophages	[87]
let-7e, miR-99b, miR-100, miR-125a, miR-125b, miR-146a, miR-146b, miR-155	Tissues, Blood Monocytes, Plasma	Protumoral Activity by Converting Monocytes Into MDSC. Reduced Response to PD-1 and CTLA-4 Inhibitors. Reduced Response to Ipilimumab and Nivolumab	[46]

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
