# Peer review of "MicroRNAs as Key Players in Melanoma Cell Resistance to MAPK and Immune Checkpoint Inhibitors"

_ijms, 2020, doi:10.3390/ijms21124544_

Round 1

Reviewer 1 Report

This manuscript is a timely review on the role of miRNA in melanoma cell resistance to MAPK and immune checkpoint inhibitors. This review summarizes the role of dysregulated miRNAs acting as oncogenes or tumor suppressors to sustain or prevent melanoma cell resistance to BRAFi/MEKi and immune therapies. The subject is important, interesting and well presented. This topic is of interest to the readership of IJMS.

It would be noteworthy to indicate in the tables whether those miRNAs have been identified solely in cell lines or also in tissues. The authors could also indicate whether  any of these miRNAs have been detected in liquid biopsies to exemplify the fact that they could be used as biomarkers of early response to treatment. A few spelling mistakes need to be corrected.

Author Response

Responses to Reviewer 1

Reviewer 1

Comments and Suggestions for Authors T

his manuscript is a timely review on the role of miRNA in melanoma cell resistance to MAPK and immune checkpoint inhibitors. This review summarizes the role of dysregulated miRNAs acting as oncogenes or tumor suppressors to sustain or prevent melanoma cell resistance to BRAFi/MEKi and immune therapies. The subject is important, interesting and well presented. This topic is of interest to the readership of IJMS. It would be noteworthy to indicate in the tables whether those miRNAs have been identified solely in cell lines or also in tissues. The authors could also indicate whether any of these miRNAs have been detected in liquid biopsies to exemplify the fact that they could be used as biomarkers of early response to treatment.

Response: we thank the Reviewer for these suggestions. In the revised Manuscript, the first required information has been included in Table 1 and Table 2. Furthermore, a new Section entitled “Predictive value of circulating miRNAs for monitoring melanoma responsiveness to targeted and immune therapies”, concerning miRNAs proposed as biomarkers of early response to treatments has been included on pages 12-13. The entire manuscript has been revised to correct typos and inaccuracies.

Reviewer 2 Report

Journal IJMS (ISSN 1422-0067) Manuscript ID ijms-836222

Title MicroRNAs as key players in melanoma cell resistance to MAPK and immune checkpoint inhibitors

Authors Maria Letizia Motti et al,

Comments:

The authors have made substantial effort to write this review manuscript by providing comprehensive details about the role of signature microRNAs in the melanoma with respect to MAPK and immune checkpoint inhibitors. Eventually the microRNAs are playing the vital role in the disease therapeutic modality specific to the therapeutic resistance and drug responsiveness. 

In my point of view the following concern needs to be address:

  1. At page -2, first paragraph, the authors should describe the potential and/or specific drug candidates subjected to the therapeutic resistance due to heterogeneity and phenotypic plasticity of melanoma.
  2. The authors stated that some miRNAS have been identified as potential biomarkers for early melanoma detection, so the list of microRNAs as signature biomarkers with its available reference range in normal and melanoma subjects as well for the resistance cases should be listed in the manuscript.
  3. Can the author provide the details about the targeted transmembrane oncogene related microRNA profile with their extracellular domain specificity, which can be a powerful tool for the identification and establishments of the soluble biomarkers.
  4. At the first sentence of the introduction the authors should provide more latest reference for the incidence and prevalence of melanoma instead of year 2014.

Author Response

Responses to Reviewers Reviewer 2

Comments and Suggestions for Authors

The authors have made substantial effort to write this review manuscript by providing comprehensive details about the role of signature microRNAs in the melanoma with respect to MAPK and immune checkpoint inhibitors. Eventually the microRNAs are playing the vital role in the disease therapeutic modality specific to the therapeutic resistance and drug responsiveness. In my point of view the following concern needs to be address:

1. At page -2, first paragraph, the authors should describe the potential and/or specific drug candidates subjected to the therapeutic resistance due to heterogeneity and phenotypic plasticity of melanoma.

Response: in the revised Manuscript, this information has been included in the Introduction on page 2 as follows: “In the past years, either dabrafenib or vemurafenib BRAF inhibitors (BRAFi) showed encouraging response rates, although the duration of response appeared to be limited [7,8]. BRAF inhibitor resistance depends on oncogenic signaling through reactivation of MAPK/Erk or activation of PI3K/Akt, which may be acquired by directly affecting genes in each pathway, by upregulation of receptor tyrosine kinases, or by affecting downstream signaling [Luebker SA and Koepsell SA (2019) Diverse Mechanisms of BRAF Inhibitor Resistance in Melanoma Identified in Clinical and Preclinical Studies. Front. Oncol. 9:268. doi: 10.3389/fonc.2019.00268]. Thus, combination of dabrafenib with the MEK inhibitor (MEKi), trametinib become worldwide employed for the care of patients with BRAF-mutant metastatic melanoma, improving their progression-free and overall survival [Long GV, Stroyakovskiy D, Gogas H, Levchenko E, de Braud F, Larkin J, et al. Dabrafenib and trametinib versus dabrafenib and placebo for Val600 BRAF-mutant melanoma: a multicentre, double-blind, phase 3 randomised controlled trial. Lancet. (2015) 386:444–51. doi: 10.1016/S0140-6736(15)60898-4, 7]. Unfortunately, patients treated with dabrafenib/trametinib combination therapy also develop alterations in the same genes that support single-agent resistance including MEK1/2 mutations, BRAF amplification, BRAF alternative splicing, and NRAS mutations [Wagle N, Van Allen EM, Treacy DJ, Frederick DT, Cooper ZA, Taylor-Weiner A, et al. MAP kinase pathway alterations in BRAF-mutant melanoma patients with acquired resistance to combined RAF/MEK inhibition. Cancer Discov. (2014) 4:61–8. doi: 10.1158/2159-8290.CD-13-0631. Long GV, Fung C, Menzies AM, Pupo GM, Carlino MS, Hyman J, et al. Increased MAPK reactivation in early resistance to dabrafenib/trametinib combination therapy of BRAF-mutant metastatic melanoma. Nat Commun. (2014) 5:5694. doi: 10.1038/ncomms6694]”. The limiting factor…… “.

2. The authors stated that some miRNAS have been identified as potential biomarkers for early melanoma detection, so the list of microRNAs as signature biomarkers with its available reference range in normal and melanoma subjects as well for the resistance cases should be listed in the manuscript.

Response: we thank the Reviewer for this suggestion. In the revised Manuscript, a new Section entitled “Predictive value of circulating miRNAs for monitoring melanoma responsiveness to targeted and immune therapies”, concerning miRNAs proposed as biomarkers of early response to treatments has been included on pages 12-13.

3. Can the author provide the details about the targeted transmembrane oncogene related microRNA profile with their extracellular domain specificity, which can be a powerful tool for the identification and establishments of the soluble biomarkers.

Response: in truth, we had some difficulty to enter the suggested topic. In the revised version of the manuscript, the following sentences have been included: in the Introduction, on page 4:"Furthermore, some microRNAs are related to the expression of transmembrane oncogenes, acting directly on their expression (e.g. EGFR) [32], or acting indirectly, by regulating the expression of soluble ligands that recognize specifically to the extracellular domain of the receptors [33]". In the Section miRNAs involved in the regulation of melanoma MAPKi-resistance on page 5 as follows:" In this regard, miR-7 could inhibit the activation of the MAPK and PI3K/AKT pathways and reverse melanoma cell resistance to BRAFi, by decreasing the expression levels of EGFR and IGF-1R."

4. At the first sentence of the introduction the authors should provide more latest reference for the incidence and prevalence of melanoma instead of year 2014.

Response: The following two more recent references have been added: Karimkhani C, Green AC, Nijsten T, Weinstock MA, Dellavalle RP, Naghavi M, Fitzmaurice C. The global burden of melanoma: results from the Global Burden of Disease Study 2015. Br J Dermatol. 2017 Jul;177(1):134-140. doi: 10.1111/bjd.15510. Epub 2017 Jun 12. PMID: 28369739; PMCID: PMC5575560. Matthews NH, Li WQ, Qureshi AA, Weinstock MA, Cho E. Epidemiology of Melanoma. In: Ward WH, Farma JM, editors. Cutaneous Melanoma: Etiology and Therapy [Internet]. Brisbane (AU): Codon Publications; 2017 Dec 21. Chapter 1. Available from: https://www.ncbi.nlm.nih.gov/books/NBK481862/ doi:10.15586/codon.cutaneousmelanoma.2017.ch1. Finally, the entire manuscript has been revised to correct typos and inaccuracies.